# Immersive and Non-Immersive Virtual Reality for Pain and Anxiety Management in Pediatric Patients with Hematological or Solid Cancer: A Systematic Review

**DOI:** 10.3390/cancers15030985

**Published:** 2023-02-03

**Authors:** Dania Comparcini, Valentina Simonetti, Francesco Galli, Ilaria Saltarella, Concetta Altamura, Marco Tomietto, Jean-François Desaphy, Giancarlo Cicolini

**Affiliations:** 1Degree Course in Nursing, Faculty of Medicine and Surgery, Politecnica delle Marche University, 60121 Ancona, Italy; 2Department of Medicine and Surgery, LUM University, 70010 Casamassima, Italy; 3Section of Pharmacology, Department of Precision and Regenerative Medicine and Ionian Area, School of Medicine, University of Bari Aldo Moro, 70124 Bari, Italy; 4Interdepartmental Center for Research in Telemedicine, University of Bari Aldo Moro, 70124 Bari, Italy; 5Department of Nursing, Midwifery and Health, Faculty of Health and Life Sciences, Northumbria University, Newcastle upon Tyne NE1 8ST, UK; 6Section of Nursing, Department of Precision and Regenerative Medicine and Ionian Area, School of Medicine, University of Bari Aldo Moro, 70124 Bari, Italy

**Keywords:** virtual reality, pain, anxiety, pediatric, cancer

## Abstract

**Simple Summary:**

Pediatric patients with cancer are often subjected to several invasive procedures and treatments, which may determine pain and induce feelings of anxiety and worry. Inadequate management of pain and anxiety can affect the overall quality of life, affecting the physical and psychological well-being of children and their caregivers. The aim of this systematic review is to summarize the findings of published experimental and quasi-experimental studies on the effectiveness of virtual reality for the management of pain and/or anxiety in children and adolescents with cancer.

**Abstract:**

Invasive and painful procedures, which often induce feelings of anxiety, are necessary components of pediatric cancer treatment, and adequate pain and anxiety management during these treatments is of pivotal importance. In this context, it is widely recognized that a holistic approach, including pharmacological and non-pharmacological modalities, such as distraction techniques, should be the standard of care. Recent evidence suggested the use of virtual reality (VR) as an effective non-pharmacological intervention in pediatrics. Therefore, this systematic review aims to analyze previously published studies on the effectiveness of VR for the management of pain and/or anxiety in children and adolescents with hematological or solid cancer. Medline, SCOPUS, Web of Science, ProQuest, CINAHL, and The Cochrane Central Register of Controlled Trials were used to search for relevant studies in accordance with the Preferred Reporting Items for Systematic Reviews and Meta-Analyses checklist. Randomized controlled trial, crossover trial, cluster randomized trial, and quasi-experimental studies were included. Thirteen studies, published between 1999 and 2022, that fulfilled the inclusion criteria were included. Regarding the primary outcomes measured, pain was considered in five studies, anxiety in three studies, and the remaining five studies analyzed the effectiveness of VR for both pain and anxiety reduction. Our findings suggested a beneficial effect of VR during painful vascular access procedures. Limited data are available on the reduction of anxiety in children with cancer.

## 1. Introduction

It is estimated that 400,000 children and adolescents of 0–19 years old are diagnosed with cancer worldwide each year, making this disease the second leading cause of death in children under the age of 15 and the fourth leading cause of death in teens aged 15 to 19 [1]. The introduction of new therapeutic protocols has significantly increased the survival rates of child and adolescent cancer patients, leading to the need to improve their long-term quality of life outcomes [2,3].

Children and adolescents with cancer, together with their caregivers, have to control several stressful factors, such as the awareness of cancer diagnosis, the disruption of their daily routines after hospitalization, and illness. Pediatric patients with cancer are subjected to several invasive procedures and treatment, which may determine pain and induce feelings of anxiety and worry. All these factors can negatively impact the overall quality of life, affecting physical and psychological well-being [4,5].

### 1.1. Background

Cancer pain is usually subdivided into chronic and acute pain. The 11th revision of the International Classification of Diseases (ICD-11) further subclassified the chronic cancer-related pain in “chronic cancer pain” and “chronic post-cancer treatment pain” [6]. The former stems from factors directly related to the cancer involvement, including metastasis, and the later refers to painful experiences from treatment interventions and procedures (procedural pain), such as venipuncture, lumbar punctures, and port access [7,8,9]. Oncologic patients are often exposed to prolonged or repeated pain whose timing and quality are influenced by the type of cancer and treatment [8]. In the pediatric setting, several studies have shown that the primary concern for patients is procedural pain, which even outweighs concerns/worries about cancer-related pain [9].

Previous experiences of procedural pain are associated with the level of anxiety and distress suffered by pediatric patients with hematological or solid cancers during their subsequent experiences [10]. In addition to physical manifestations, painful experiences may also affect psychological aspects causing/determining anxiety in both adults and children with cancer [8]. Anxiety is defined as a psychological condition associated with intense worry and/or fear in response to specific environmental stimuli, in the absence of a proper adaptive reaction [11]. In turn, anxiety may worsen pain sensations leading the patients to avoid situations and places that have given rise to these sensations [12]. In this context, failure to manage pain and anxiety can increase the risk of adverse events, including post-traumatic stress disorder, maladaptive behaviors [13], altered pain perception, development of chronic pain, and an increased sensitivity to future painful stimuli. Furthermore, this can increase the risk of health care avoidance behaviors [14], including a reduced compliance with oncological procedures and, therefore, the worsening of treatment outcomes.

It is widely recognized that a comprehensive and holistic approach, including pharmacological and non-pharmacological modalities, for the treatment of cancer pain should be the standard of care [15]. Pharmacological therapies, particularly non-steroidal anti-inflammatory drugs and opioids, represent the cornerstone of pain management in pediatric patients with cancer [16,17]. However common side effects may occur, such as nausea, vomiting, constipation, and respiratory depression, which can require longer hospitalization (LOS) in inpatient settings, increasing healthcare costs and decreasing patient satisfaction.

Research in the past years has focused on the development of effective active and passive distraction methods for pain and anxiety management in children and adolescents, which may easily be integrated into medical and nursing care [18]. Specifically, in oncological settings, these distraction methods include the use of music, massage, breathing exercises, hypnosis, and behavioral therapy [4,19,20,21,22]. However, distraction interventions are often poorly implemented because they are time-consuming and require adequate training. An innovative method of active distraction for overcoming stressful or painful medical procedures is the Virtual Reality (VR), a digital simulation of a computer-generated situation or environment, in which hospitalized patients may feel immersed and interact with three-dimensional objects [14]. VR can be either immersive (IVR) or non-immersive depending on the degree of viewer isolation from the surrounding physical environment when interacting with the virtual scene. Non-immersive VR implements the virtual scene using devices with traditional graphics, such as a large display or wall screen, whereas IVR commonly take advantage of head-mounted display/visor and motion-tracking systems to provide a full immersion and interaction with the virtual environment [23]. The main difference between immersive and non-immersive VR is the participant’s point-of-view and the experience produced during use. Through IVR, participants experience a great sense of presence inside the environment and can view the full panorama, while non-immersive VR only allow participants to see the contents based on how the device in use—PC, smartphone, or tablet—is held and moved [24,25].

Due to the current technological evolution and children’s ability to interact with these devices, recent evidence suggested the use of VR as an effective non-pharmacological intervention in pediatrics, because the inclusion of “gamification” (game elements in educational context) allows children a distraction due to their immersion in virtual play [14,26,27,28,29,30,31,32]. Actually, VR was tested as an effective method in the management of pain and anxiety of children in oncologic, medical, surgical, and community settings (i.e., during vaccination procedures) [14,23,27,30]. The significant impact of VR on supportive care management and quality of life enhancement of pediatric patients with cancer is well established [9,32,33].

Two systematic reviews published in 2018 evaluated the results of studies focused on the use of VR for pain and anxiety management in both adults and pediatric oncologic patients [34,35]. Thus, there is no comprehensive updated systematic review, including the last 4 years, and specifically assessing the effectiveness of VR on pain and anxiety management in pediatric patients with hematological or solid cancer.

### 1.2. Objectives

The main objective of this systematic review was to assess the effectiveness of VR for the management of pain and/or anxiety in children and adolescents with hematological or solid cancer.

## 2. Methods

### 2.1. Design

A systematic review of the literature was performed according to the methods outlined in the Cochrane Handbook for Systematic Reviews of Interventions [36]. The results were reported as prescribed by the Preferred Reporting Items for Systematic Reviews and Meta-Analyses checklist (PRISMA) [37]. The study protocol was registered with the PROSPERO register, number (blinded for Referee), available at (blinded for Referee).

### 2.2. Search Strategy

#### 2.2.1. Electronic Searches

A systematic search was performed to identify clinical trials of VR for pain and anxiety management using the following electronic databases up to September 2022: Medline (through PubMed), SCOPUS, Web of Science, ProQuest, CINAHL (through (EbscoHOST), The Cochrane Central Register of Controlled Trials (CENTRAL) via CRSO.

A combination of controlled vocabulary (i.e., Medical Subject Headings—MeSH) and free text terms were used. The main terms included were “virtual reality”, “pain”, “cancer pain”, “anxiety”, “cancer anxiety”, “pediatric oncologic patients”, “cancer”, “child*”, “adolescent”. Table 1 shows a description of search strategies for each database. The publication languages were restricted to English and Italian.

#### 2.2.2. Searching Other Resources

In addition to the electronic searches, we searched the following registers for ongoing trials: Clinical trials register (Clinicaltrials.gov, accessed on 2 January 2023) and ISRCTN registry (https://www.isrctn.com, accessed on 2 January 2023). The references list from retrieved eligible studies were also reviewed for other articles not retrieved in the initial search and were potentially eligible for inclusion.

### 2.3. Inclusion and Exclusion Criteria

The inclusion criteria were based on the Participants, Intervention, Comparator, and Outcomes (PICO) [38] and are reported in Table 2.

Population: pediatric patients included in the meta-analysis were pre-school (4–5 years) and school (6–12) children and adolescents (13–19) diagnosed with a hematological or solid tumor, including both inpatient and outpatient settings, who had undergone painful and/or anxiogenic medical and nursing procedures or cancer treatments. Medical or nursing procedures refer, for example, to vascular access procedures, needle insertion (i.e., port access procedures), blood samples, bone marrow aspiration, lumbar punctures, and arterial punctures. Cancer treatments include all the procedures related to chemotherapy, radiotherapy, and surgery.

Intervention(s): VR interventions of any intensity or duration, implemented with or without pharmacological support, to manage pain and/or anxiety in pediatric cancer patients undergoing medical procedures or cancer treatments, in order to offer a comprehensive view of both treatments’ effect. In particular: (1) IVR intervention characterized by the blocking of the view on the external environment (i.e., through a helmet-mounted display-based systems and projection VR systems), which determines the immersion of the patient in a three-dimensional virtual environment; (2) non-IVR, in which patients interact with a scenario displayed on a mobile phone, tablet, three-dimensional glasses, or a computer, without being completely immersed and, together with the digital images, can always perceive the real world around them.

Comparator(s): VR compared to: (i) no distraction or standard of care/usual care; (ii) non-VR distraction (i.e., videogames, television, music, massage, breathing exercises, hypnosis, and behavioral therapy); (iii) other digital technology distractors (i.e., socially assistive robots, including humanoid and pet robots).

Outcome(s): pain and anxiety, measured using self-reported or observer-reported measurements or both.

Types of studies: randomized controlled trials (RCTs), as well as crossover trials and cluster randomized trials, and quasi-experimental studies.

Studies that met the following criteria were excluded: (i) carried out in adult population or pediatric and adolescent patients without a diagnosis of cancer; (ii) including pediatric patients of other age groups (e.g., children aged 0–3 years); (iii) investigating the effects of virtual reality in which pain and anxiety were not primary or secondary outcomes; (iv) studies including outcomes different from perceived pain (i.e., heart rate as a physiological measure of arousal) and/or anxiety (i.e., studies reported exclusively measures of fear, maladaptive behavior, or distress); (v) observational studies, qualitative studies, reviews and meta-analyses, editorials, commentaries, letters to Editor, conference papers, abstracts, dissertations, case-studies, case-series studies, and quasi-experimental studies without a control group design (Table 2).

### 2.4. Screening

Search results were assessed for a first screening from one reviewer (DC), then following removal of duplicates, two reviewers (DC and VS) independently screened the remaining studies against the inclusion criteria based on the title and abstract of potentially relevant articles, followed by the remaining articles based on the full text. Disagreements relating to article inclusion were resolved through a discussion with a third author (GC or MT) to reach a final consensus.

The selection of studies was conducted through an initial screening of the title and abstract to identify potentially relevant articles. Then, a screening was carried out of all the full text articles identified as relevant in the initial selection. Additional papers, not identified in the initial literature search, were obtained through the examination of the references in the published studies.

### 2.5. Quality Assessment

Quality appraisal for the included RCTs were conducted by two reviewers (DC and VS) using “The revised Cochrane risk-of-bias tool for randomized trials (RoB2)” [39] for RCTs. This tool considers five domains: bias arising from randomization process, bias due to deviations from intended interventions, bias due to missing outcome data, bias in the measurement of the outcome, bias in the selection of the reported result.

The “Risk of Bias in Non-randomized Studies—of Interventions (ROBINS-I)” tool [40] was used to conduct the quality appraisal of quasi-experimental studies. This tool covers seven key domains of bias: (i) due to confounding, (ii) in selection of participants, (iii) in classification of interventions, (iv) due to deviations from intended interventions, (v) due to missing data, (vi) in measurement of outcomes, and (vii) in selection of the reported results. The risk of bias (RoB) was assessed using the following scoring: low, moderate, serious, or critical risk of bias and “No information” on which to base a judgement of RoB if no clear indication exists to indicate that the study is at serious or critical RoB and if there is a lack of information in one or more key domains of bias.

### 2.6. Data Extraction

To record the key information from the included studies, a chart form was developed to categorize the included studies’ key features. One reviewer (DC) extracted data from the included studies by using a screening form based on the predetermined inclusion criteria, then a second reviewer (VS) validated the extracted data. The following main information were extracted from the included studies: study author(s), year of publication, study design, participant characteristics, intervention and comparison groups, outcomes (pain and anxiety) and measures, procedures and treatments, type of VR (equipment, software, VR environment and applications), and key results.

## 3. Results

### 3.1. Study Selection

The database search strategies yielded 2188 records, and 1969 duplicates were excluded. Furthermore, 202 studies were excluded after screening the titles and abstracts. Three additional studies were excluded because they did not meet the inclusion criteria [41,42,43] or reported insufficient data [44]. Thus, 13 studies were finally included in the systematic review (Figure 1).

#### Ongoing Studies

A search of ongoing trials on the ClinicalTrials.gov (accessed on 2 January 2023) site and on the ISRCTN registry showed ten studies that are investigating the use of VR to manage pain and anxiety in pediatric patients with cancer (Table 3).

### 3.2. Quality Assessment

Regarding the quality assessment of the included RCT, two studies [45,46] reached an overall “low” RoB judgement. However, in the evaluation of the randomization process, only the study of Wong and collaborators [46] clearly specified that the research assistant and patients were blinded to group allocation until baseline data collection was completed. For the remaining studies, in the randomization process domain, the judgement was “some concerns” for the RoB. However, given the nature of the intervention, neither the patients nor the researcher blinding was applicable, and this aspect has been appropriately evaluated in the randomization process domain for each RCT.

For three of the seven RCTs, the quality assessment overall judgement was “some concerns” for the RoB because in these studies, insufficient data were provided on the process of random sequence generation also [47,48,49]. In their study, Wolitsky and collaborators specified that two participants withdrew but did not specify the time point of withdrawing [49]; however, we rated this study as “low” RoB for the domain “missing data outcome” because data missing were less than 10%.

Two studies were assessed as a “high” overall RoB for bias arising from the randomization process and bias due to missing outcome data (more than 10%) [50,51].

Regarding the assessment of RoB in the remaining studies using the ROBINS-I tool, all studies were classified as have a “moderate” RoB [52,53,54,55,56,57].

### 3.3. Characteristics of Included Studies

Table 4 and Table 5 show the main characteristics of included studies. This review included five randomized clinical trials [46,47,49,50,51], two pilot RCT [45,48], one experimental crossover design [54], one experimental control group design [55], two quasi-experimental pretest–post-test between-subjects design [52,56], one intervention-comparison group parallel group design [53], and one interrupted time series design [57].

The included studies were published between 1999 and 2022; seven of them (54%) after 2018. Most studies were carried out in the United States (three studies), followed by China, Turkey, and Italy (two studies), and Australia, Canada, Iran, and Sweden (one study).

#### 3.3.1. Participants and Procedures

Overall, 644 children and adolescents with hematological or solid cancer were involved. All included studies analyzed pediatric patients with active cancer; only two studies mixed children with cancer and participants with unspecified hematological diseases [50,53]. However, as reported in the inclusion criteria of the review (Table 2), in these studies more than 70% of the sample was diagnosed with hematological or solid cancer. Therefore, we decided to include it even if data for patients with cancer could not be isolated. Five of the included studies did not specify the cancer diagnosis of participants [47,49,50,51,54].

Regarding the painful and anxiogenic medical or nursing procedures for which VR was applied, most of the studies considered the Huber needle insertion into a port access [47,48,49,51], both port procedures and venipunctures [53], or exclusively venipunctures [50]. In three studies, VR was applied for children undergoing active/chemotherapy treatment [52,56,57]. Other studies considered the lumbar punctures [55], peripheral venous cannulation [46], and central venous catheter (CVC) dressing [54]. In addition, one study included the experience of the inpatient oncology admission as an anxiety-inducing event [45].

#### 3.3.2. Outcomes and Measures

Regarding the primary outcomes measured (pain and/or anxiety), pain was considered in five studies [48,50,51,53,55] and anxiety in three studies [52,54,57]. The remaining studies analyzed the effectiveness of VR for both pain and anxiety [45,46,47,49,56]. Five studies measured pain using the Visual Analogue Scale (VAS), while the remaining studies used the Wong–Baker FACES (WBF) Pain Rating Scale, the Numeric Rating Scale (NRS), the McGill Pain Questionnaire (MPQ), the Children’s Hospital of Eastern Ontario Pain Scale (CHEOPS), the Face, Legs, Activity, Cry, Consolability Scale (FLACC), or the Color Analogue Scale (CAS).

Anxiety was assessed using the Children’s Anxiety Meter-State (CAM-S), the short form of the Chinese version of State Anxiety Scale for Children (CSAS-C), the Revised Children’s Manifest Anxiety Scale (RCMAS-2), the short version of the Pain Anxiety Symptom Scale (PASS-20), the Visual Analogue Scale (VAS), the State-Trait Anxiety Inventory for Children (STAIC-1), and the Facial Affective Scale (FAS) (Table 4). Table 6 shows the secondary outcomes measured in the included studies.

### 3.4. Intervention, Comparators, and Type of VR

Two studies compared a VR intervention group to a non-immersive iPad distraction [45,48], whereas the remaining studies compared a VR intervention group to the standard of care with no distraction, although the standard of care was not well defined in each study. In addition, the description of the control condition was not clearly stated in two studies, in which the control condition was defined as “no VR treatment” [49] or “no intervention” [56] without adding details.

Regarding the type of VR used in the different procedures, most studies implemented immersive VR interventions through an immersive head-mounted display. Only two studies used non-immersive VR to evaluate outcomes; specifically, Li and collaborators projected play spaces in a dedicated room to explore the effectiveness of VR in reducing children’s anxiety [52]; Nilsson and collaborators used a computer screen displaying virtual world games to evaluate the effectiveness of VR in pain management [53]. Studies were heterogeneous regarding VR equipment and environmental software (Table 4).

### 3.5. Effect of VR on Pain

Overall, ten studies analyzed the effectiveness of VR (immersive or non-immersive) in the management of children’s pain (Table 4). One study compared non-immersive VR to no distraction (standard of care) [53]; seven studies compared immersive VR to no distraction (standard of care) [46,47,49,50,51,55,56]; two studies compared immersive VR to non-VR distraction(s) [45,48].

#### 3.5.1. Summary of Main Results Based on the Type of VR

Overall, for pain intensity during various procedures assessed in the included studies, we found no statistically significant effect for interventions of non-IVR compared to IVR or no distraction/standard of care, both in self-reported and observed patient pain. Whereas we found evidence of a beneficial effect of IVR compared to no distraction/standard of care for self-reported, parent-reported, and observed pain.

#### 3.5.2. Non-Immersive VR Compared to the Standard of Care

The comparison between non-immersive VR and the standard of care was reported in a quasi-experimental study carried out to explore the effect of VR during a needle-related procedure [53]. This study evaluated both the patients’ self-reported pain using the CAS scale and the nurse-observed pain using the FLACC scale (during and post-procedure). Overall, the study found no evidence of a beneficial effect of VR on pain; findings showed no statistically significant differences between the VR group and the control group. However, the analyses of the FLACC scores before and during the procedure showed that the scores did not increase in the intervention group compared to the control one.

#### 3.5.3. Immersive VR Compared to the Standard of Care

##### Pain: Self-Report

Four studies analyzed the efficacy of VR through the assessment of patients’ self-reported pain [46,50,55,56]. Among these, three studies demonstrated a statistically significant reduction in pain scores in the intervention group compared to the control group [46,50,56].

A within-subjects crossover RCT assessed pain associated to a venipuncture procedure (during and post procedure) by using the VAS scale [50]. The results showed a significantly lower pain level during the VR intervention compared with the pain level during the “No VR” intervention. Another study employed a quasi-experimental pretest–post-test design (at 7-day and 1-month pretest–post-test intervals) and explored the effect of VR on adolescents’ pain during a chemotherapy treatment using the McGill Pain questionnaire [56]. Findings showed a significant difference in the pain scores in the stages of post-test, first follow-up, and second follow-up between the control group and the experimental group (*p* ˂ 0.01). These results indicated that VR had a beneficial effect on pain and that this effect remained constant in the first and second follow-up periods. In this study, neither the type of chemotherapy side effects determining pain, nor the specific type of treatment were detailed.

A RCT assessed the effectiveness of VR in reducing pain in patients undergoing peripheral intravenous cannulation (PIC) (immediately after the procedure) by using the VAS scale [46]. After normalization to the number of previous PICs received by the patients, results showed a smaller increase in pain score after PIC in the VR group compared to the control group (estimated mean difference = −1.69, *p* = 0.007). The difference in the changes in pain levels between the intervention and control group was significant only among patients aged 12 to 17 years (estimated mean difference = −2.20 (95% CI: −4.24, −0.16), *p* = 0.03).

One pilot study using an experimental control group design to assess the effect of VR on adolescents undergoing a lumbar puncture using the VAS scale found a lower pain score in the VR group compared to the control group [55]. However, differences between the two groups were not statistically significant (*p* = 0.77).

##### Pain: Self-Report and Parental Report

Two studies analyzed both the patients’ self-reported pain and the parental reported pain using the BWF scale (during and post-procedure) and found evidence for the effectiveness of VR in reducing pain during the venous port access procedure [47,51]. According to the self and parent-reported pain scores, these studies showed lower pain scores in the VR group compared to the control group with a statistically significant difference between groups (*p* < 0.05).

##### Pain: Self-Report and Observer

One study explored the effectiveness of VR in reducing pain in children undergoing a port access procedure (before and during the procedure) using the VAS and the CHEOPS scales as measures of self-reported and observed pain, respectively [49]. The authors considered a composite measure of distress before and during the procedure, consisting of distress before the procedure, measured as the mean of the child’s self-report of anxiety and predicted pain, and distress-during the procedure, calculated as the mean of the child’s retrospective self-report of pain and anxiety during the procedure. This study highlighted a positive effect of VR intervention compared to no distraction during the procedure, only for the observer-reported pain (*p* < 0.01) but not for patients’ self-reported pain (*p* = 0.10).

#### 3.5.4. Immersive VR Compared to Non-Immersive Distraction and Standard of Care

Both the included studies for the comparison between immersive VR and non-immersive distraction found no statistically significant difference between study’s groups (Table 3 and Table 4).

In a pilot RCT, immersive VR was compared to a non-immersive iPad distraction (children watched a video while wearing the headphones) during a port access procedure. The standard of care included parental presence, the use of a topical anesthetic, and child life specialist involvement. In this study, pain was assessed with the NRS scale through the children’s self-report and through reports of parents, nurses, and researcher observers; preliminary results on the efficacy of immersive VR vs. non-immersive distraction were based only on the participant’s self-reporting perceptions. Lower pain was reported in the VR group compared with the iPad group. Overall, more participants reported no pain in the VR group (65%) than in the iPad group (45%) during the procedure. However, no statistically significant difference between groups was reached [48].

A single-blind RCT compared VR to a non-immersive iPad distraction (children watched the identical video presented in the VR condition) during an inpatient oncology admission [45]. This study measured the effects of intervention on common cancer- related emotional and physical distress symptoms, including pain. The VR intervention was delivered by research assistants at the patients’ beside (undergoing treatment or procedures were not specified). Self-reported pain was assessed using the VAS scale. No statistically significant difference was found between conditions (VR vs. iPad) on pain, but findings showed a greater ability of VR to alleviate pain’s post-treatment scores than the iPad. Specifically, pain scores were lower (*p* = 0.056) in VR condition (m = 8.19, se = 2.36) compared to the iPad group (m = 15.52, se = 2.92) for females [43].

### 3.6. Effect of VR on Anxiety

Overall, nine studies analyzed the effectiveness of VR (immersive or non-immersive) in the management of children’s anxiety (Table 4).

One study compared non-immersive VR to standard of care [52]. Six studies compared immersive VR to standard of care [46,47,49,54,56,57]. Finally, one study compared immersive VR to non-immersive distraction(s) [45].

#### 3.6.1. Summary of Main Results Based on the Type of VR

Overall, for the management of children’s anxiety during procedures, we found insufficient and inconsistence evidence of an effect of non-IVR compared to no distraction. While we found evidence of a beneficial effect favoring IVR compared to no distraction/standard of care for both self-reported and parent reported, there was insufficient evidence to confirm the beneficial effect of IVR compared to non-IVR for the management of patients’ anxiety.

#### 3.6.2. Non-Immersive VR Compared to Standard of Care

One study explored the effectiveness of VR in the management of anxiety in patients admitted to an oncology ward for cancer treatment [52]. The CSAS-C scale was used to measure children’s self-reported levels of anxiety (total scores ranging from 10 to 30). This non-equivalent control group pretest–post-test between-subject design was divided into two phases: first, all the participants admitted to the unit received usual care (control group); second, after a one-month washout period, all the participants admitted to the unit received usual care plus therapeutic play (experimental group). Therapeutic play consisted of 30 min VR computer games daily, five days a week. The anxiety level was assessed at baseline (at the admission to the ward) and on day 7 after the admission. Children showed quite high state anxiety on admission in both groups (VR group: mean = 21.04, SD = 4.90; control group: mean = 21.11, SD = 4.66; *p* = 0.93) and a slight decrease on day 7 (VR group: mean = 19.48, SD = 4.73; control group: mean = 21.06, SD = 4.52). The results of between-subject effects showed that there was no statistically significant main effect for the intervention, with no difference in participants’ anxiety scores between the two groups on day 7 (F = 3.48, *p* = 0.07, eta squared = 0.05, observed power = 0.68).

#### 3.6.3. Immersive VR Compared to Standard of Care

##### Anxiety: Self-Report

A pilot feasibility study (experimental cross-over design) aimed at evaluating the effect of VR during CVC dressing change by comparing one VR session with another session with no VR in the same children [54]. Anxiety levels were measured with the RCMAS-2 scale. No decrease was observed in the anxiety scores of VR compared with no-VR.

Schneider and Workman carried out an interrupted time series with removed treatment design to explore if VR was an effective distractor in reducing chemotherapy-related symptoms of distress (including anxiety) in children [57]. The STAIC-1 scale was used to measure participants’ anxiety during three different chemotherapy sessions, the second of which included the VR intervention. The STAIC-1 was administered pre, post (immediately following the chemotherapy treatment) and after 48 h. The lowest mean STAIC-1 scores were recorded immediately after the treatment and 48 h following the second (VR) chemotherapy treatment. A difference in the STAIC-1 over the three time measures was found; however, difference cannot be attributed to the VR (*p* = 0.07).

The findings of Wolitsky and collaborators suggested that children in the VR group exhibited less anxiety during the procedure, but no significant difference was found between groups (*p* = 0.10) [49].

Sharifpour and collaborators carried out a quasi-experimental pre-test, post-test, between-subject design (timing: at 7-day and 1-month pretest, post-test intervals) and used the PASS instrument to assess the pain-related anxiety; findings showed that the use of VR caused a 94.9% difference in pain anxiety reduction [56]. The time-by-group interaction effect on the perceived anxiety was significant (*p* ˂ 0.01). In the stages of the post-test, first follow-up (at 7-day) and second follow-up (at 1-month), there was a significant difference in the score of anxiety between the control group and the experimental group (*p* ˂ 0.01).

In another study, the participants’ anxiety levels were measured using the CSAS-C scale [46]. The intervention group showed a significant reduction in anxiety levels (estimated mean difference = −3.50, *p* < 0.001) compared with the control group. The difference in the changes in anxiety levels between the intervention and control group was significant among patients aged 6–11 years (estimated mean difference = −3.85 (95% CI: −6.00, −1.71, *p* < 0.001) and 12–17 years (estimated mean difference = −2.90 (95% CI: −5.04, −0.76, *p* = 0.008).

##### Anxiety: Self-Report and Parental Report

Gerçeker and collaborators analyzed both the patients’ self-reported anxiety and the parental reported anxiety using the CAS-D anxiety score, during the port needle insertion [47]. This study found evidence on the effectiveness of VR (*p* < 0.001) in reducing self-reported and parent-reported anxiety scores in the VR compared to the control group (mean = 2.9, SD = 2.0; mean = 5.4, SD = 2.0, respectively).

#### 3.6.4. Immersive VR Compared to Non-Immersive Distraction and Standard of Care

In one study, anxiety was assessed using the VAS scale in children receiving VR or iPad distraction [45]. No statistically significant differences were found between groups (VR vs. iPad) on anxiety, but VR intervention was more effective in decreasing anxiety post-treatment scores than iPad conditions. Anxiety was lower in VR (m = 5.51, se = 3.02) compared to the iPad group (m = 13.99, se = 3.75, *p* = 0.083) for females.

## 4. Discussion

To the best of our knowledge, this is the first systematic review designed to examine the effectiveness of both immersive and non-immersive VR as a distraction measure to reduce pain and anxiety in pediatric patients with hematological or solid cancers. This review suggests a general trend towards the benefit of VR with respect to standard of care or to other types of non-immersive distractions used to manage children’s pain and anxiety in various cancer-related procedures. These results are in line with a review of studies performed on a mixed population (adults and pediatrics) [34].

Overall, lower scores in the VR group compared to the control group were self-reported by children for pain [45,46,47,48,49,50,51,53,55,56] and anxiety [44,45,46,47,52,54,57] or were reported by parents for both pain and anxiety [47,48,51]. A few studies also reported positive findings of observed outcomes [48,49,53]. However, the statistical significance of VR benefits was shown in five out of ten included studies that evaluated children’s pain and in three out of eight studies that evaluated children’s anxiety. These results further support the conclusion drawn from the recent meta-analysis of Cheng and collaborators [58] that encouraged VR as a distraction in the clinical care of children with cancer, because it can lessen pain and anxiety in this population. Moreover, authors showed that immersive VR can relieve fear of painful medical procedures in children with cancer, hinting at the potential benefits of VR.

### 4.1. Outcomes Measured and Application

Regarding the outcomes measured, most studies assessed pain, while few studies analyzed the effect of VR on procedural anxiety. Although the results on the effectiveness of VR in reducing anxiety appeared encouraging, our review suggests the need for further experimental studies evaluating this outcome to establish more robust evidence.

In terms of procedure for which the effect of VR was measured, it is noteworthy that all the studies that reached statistical significance compared immersive VR with standard of care to evaluate the effect of the distractor during venipuncture or vascular access procedures, particularly the port access procedure. This confirms a previous systematic review highlighting that immersive-VR may be an effective distractor method for the management of needle-related procedures in children [28]. Additionally, this result reinforces the recommendation by Loeffen and collaborators of implementing an active distractor to reduce procedural pain and distress in children with cancer [2].

However, only a single study reported statistically significant results [56]. Moreover, this study was characterized by a small sample size (30 adolescents) and limited details on variables that could have influenced the results, such as the stage of cancer, the previous experience of hospitalization and treatment sessions, and the severity of the disease. The chemotherapy treatment process is characterized by several unpleasant sensations attributable to pain from venipuncture or port access, anticipatory nausea and pretreatment anxiety, or anxiety that can arise when patients observe their own treatment or that of others [59,60,61]. Pain and anxiety triggered by a chemotherapy treatment differ from procedural pain and procedural-related anxiety and, therefore, differentially affect the effectiveness of VR as a distractor device. Regarding the non-pharmacological interventions to improve outcomes in patients with cancer, previous studies on an adult population showed that VR is a promising intervention for chemotherapy recipients [62,63,64]. Although the distress of chemotherapy treatment in children with cancer may widely affect the overall course of their treatment and quality of life, the topic is poorly addressed in the literature.

### 4.2. Type of Intervention and Comparators

Nearly all the studies that achieved statistical significance used immersive-VR systems and compared VR with no intervention and usual care. In these studies, VR was found to be more effective in reducing pain and anxiety than standard care. Nevertheless, findings of this review cannot clearly differentiate between the positive value of VR with respect to other forms of distraction or non-VR distraction because of two main reasons.

First, the standard of care was not always well-defined. For example, parental engagement was clearly stated only in a few studies [48,50], as well as non-medical conversation or instructions delivered to patients [46,47,50]. In addition, in studies investigating procedural pain, whether or which kind of pharmacological analgesia was used was not always well described.

Second, only two of the included studies compared VR intervention with a non-immersive iPad distraction added to the standard of care [45,48].

Moreover, the included studies involved various types of VR and environments; therefore, it is not possible to determine which scenario of VR could be the most effective one to reduce children pain or anxiety during different types of procedure.

### 4.3. Limits

There are some limitations in this study. First, despite the extensive search strategy performed, with no restriction on year of publication, some relevant studies published in language other than English and Italian may have been excluded. Second, the outcome measurement tools, the painful procedures among studies, the content of interventions, and the types of VR scenarios reported in the included studies were varied; therefore, due to this heterogeneity, only narrative synthesis was possible. Finally, most studies were monocentric in design and had small sample sizes; this may limit the generalizability of our findings.

## 5. Conclusions

VR distraction has the potential to be an effective distractor for the management of pain and anxiety in children with hematological or solid cancer.

Particularly, most of the included studies suggested a beneficial effect of immersive VR during painful vascular access procedures. However, an existing gap in the literature is the limited data available on the effectiveness of VR in reducing anxiety in children with cancer and, more specifically, anxiety related to chemotherapy treatments. To support clinical decisions, there is a need to establish the procedure or intervention types for which the VR distraction may be most useful and effective. For treatment that required frequent or repeated sessions of therapy, such as chemotherapy, it would be important to explore the effects of repeated/periodic/recurrent VR exposure over time.

Based on current studies, this review showed inconclusive evidence of the beneficial effect of VR in reducing pain and anxiety in children and adolescents receiving chemotherapy. Thus, it appears necessary to carry out further research with experimental designs, high methodological qualities, and larger sample sizes.

Overall, more experimental studies, with a larger sample size, considering similar conditions or procedures, and, as much as possible, using the same outcomes assessment tools, are needed to provide unequivocal evidence for the effectiveness of VR in the pediatric cancer population.

## Figures and Tables

**Figure 1 cancers-15-00985-f001:**
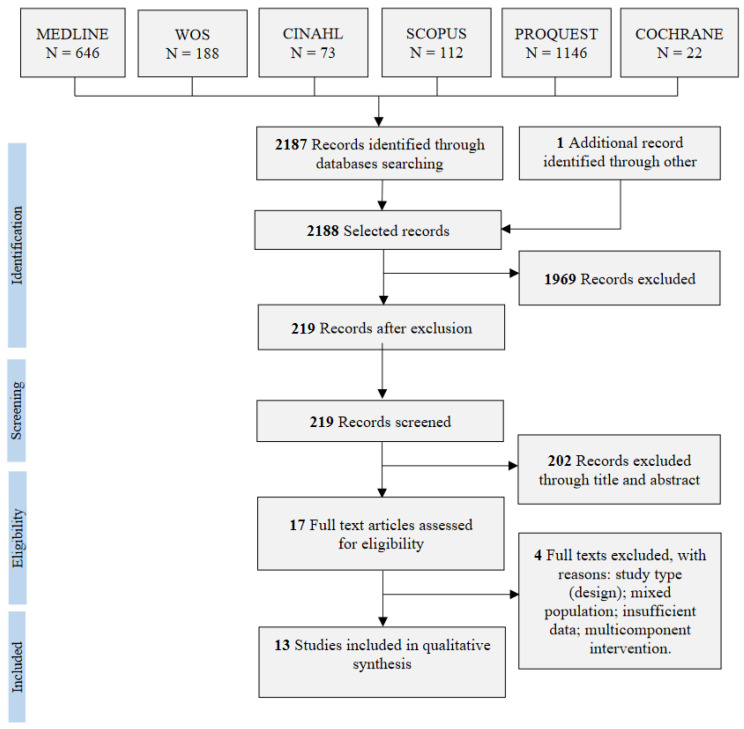
PRISMA flow diagram of study selection.

**Table 1 cancers-15-00985-t001:** Literature search of the electronic databases.

Database /Strategy#	Search	Results
Medline (PubMed)	(((((“Virtual Reality”[Mesh Terms] OR “Virtual Reality Exposure Therapy”[MeSH Terms]) AND “Hematologic Neoplasms”[MeSH Terms] AND “Neoplasms”[MeSH Terms] AND “Pain”[MeSH Terms]) OR “Pain Management”[MeSH Terms]) AND “Anxiety”[MeSH Terms]) OR ((((“Virtual Reality”[MeSH Terms] OR (“virtual”[All Fields] AND “reality”[All Fields]) OR “Virtual Reality”[All Fields]) AND “pediatric*”[All Fields] AND (“cancer s”[All Fields] OR “cancerated”[All Fields] OR “canceration”[All Fields] OR “cancerization”[All Fields] OR “cancerized”[All Fields] OR “cancerous”[All Fields] OR “Neoplasms”[MeSH Terms] OR “Neoplasms”[All Fields] OR “cancer”[All Fields] OR “cancers”[All Fields])) OR ((“child”[MeSH Terms] OR “child”[All Fields] OR “children”[All Fields] OR “child s”[All Fields] OR “children s”[All Fields] OR “childrens”[All Fields] OR “childs”[All Fields]) AND (“cancer s”[All Fields] OR “cancerated”[All Fields] OR “canceration”[All Fields] OR “cancerization”[All Fields] OR “cancerized”[All Fields] OR “cancerous”[All Fields] OR “Neoplasms”[MeSH Terms] OR “Neoplasms”[All Fields] OR “cancer”[All Fields] OR “cancers”[All Fields]) AND (“Pain”[MeSH Terms] OR “Pain”[All Fields]) AND (“Anxiety”[MeSH Terms] OR “Anxiety”[All Fields] OR “anxieties”[All Fields] OR “anxiety s”[All Fields]))) AND (“child”[MeSH Terms:noexp] OR “adolescent”[MeSH Terms] OR “child, preschool”[MeSH Terms]))) AND (preschoolchild[Filter] OR child[Filter] OR adolescent[Filter])	646
Scopus	TITLE-ABS-KEY (virtual AND reality AND cancer AND pediatric* OR child*); TITLE-ABS-KEY (virtual AND reality OR virtual AND reality AND intervention AND cancer OR oncology AND pediatric* OR child* AND pain AND anxiety)	124
CINAHL	virtual reality AND children AND (cancer patients or oncology patient s or patients with cancer); (virtual reality or vr or immersive or simulation or head mounted display) AND cancer AND pediatric	73
Medline (Pubmed)	(“mouth care” OR “mouth diagnostic” OR “oral care” OR “oral hygiene” OR “dental care” OR “dental health”) AND (“critical care” OR “intensive care” OR “ICU”) AND nurs* AND (Guideline OR effect* OR interven* OR program OR tool OR treat* OR prevent* OR train*)	253
Web of Science	virtual reality (Topic) and children (Topic) and pain (Topic) and anxiety (Topic) and Review Article or Abstract or Case Report or Letter or Book or Meeting (Exclude—Document Types)	188
ProQuest	virtual reality and pediatric* and cancer and pain and anxiety	1.146
Cochrane	virtual reality in Title Abstract Keyword AND pediatric* in Title Abstract Keyword AND “Cancer” in Title Abstract Keyword—(Word variations have been searched)	22

**Table 2 cancers-15-00985-t002:** Inclusion and exclusion criteria for eligibility in the review.

	Inclusion Criteria	Exclusion Criteria
Population	Pediatric patients: pre-school (4–5 years), school (6–12 years), and Adolescents (13–19 years). Studies of children and adults (mixed population) may be included if: - Data for children may be isolated.	Children aged 0–3 years; adults aged 20–65; elderly populations (>65).
	Males and females.	No exclusion criteria.
	Patients with a diagnosis of cancer (hematological malignancies or solid tumors) regardless of cancer type, stage, and anticancer treatment phase. Studies on cancer patients and other hematological diseases (mixed) may be included if: - Data for patients with cancer may be isolated or a minimum of 60% of the sample was diagnosed with hematological cancer or solid tumor.	No exclusion criteria.
	Patients undergoing painful and/or anxiety-inducing medical procedures/cancer treatments.	No exclusion criteria.
Intervention	Virtual Reality (Immersive or non-Immersive).	Non-VR interventions. Multicomponent interventions in which the effect of VR could not be isolated.
Comparator/ Control	Any group(s) or control group(s): non-digital technology distraction modalities; usual care/standard of care; non-Virtual Reality digital technology distraction interventions.	Studies that do not include comparative data.
Outcomes	Pain and anxiety based on behavioural observations and self-reports (from patients, parents, healthcare workers, and researchers).	Outcomes different from pain (i.e., studies reported exclusively heart rate as a physiological measure of arousal) and anxiety (i.e., studies reported exclusively measures of fear, maladaptive behavior, or distress).
Setting	Any geographical location, any cultural factors (e.g., race/ethnicity, gender), and any healthcare settings, including inpatients and outpatient’s settings.	No exclusion criteria.
Study type	Experimental or quasi-experimental studies.	Observational studies, qualitative studies, review, editorial, commentary, letter to Editor, conference paper, abstract, dissertations, case-study, case-series studies, and quasi-experimental studies without a control/comparison group.
Additional criteria	Peer-reviewed or pre-printed studies.	Non-peer-reviewed studies. Grey literature.
	Articles published in English or Italian.	Studies published in any language other than English and Italian.

**Table 3 cancers-15-00985-t003:** Ongoing studies investigating the effectiveness of VR in the management of pain and anxiety in pediatric cancer patients.

Number/ Identified	Recruitment Status	Study Completion Date	Study Title	Design, Participants	Application	Outcomes	
						Pain (Y/N)	Anxiety (Y/N)
NCT03435367	Completed	Missed information	Immersive Virtual Reality to Reduce Procedural Pain During IV Insertion in Children in the Emergency Department: A Feasibility Pilot Study	Missed information	Missed information	-	-
NCT03888690	Unknown status	Estimated Primary Completion Date: 31 March 2020	Randomized Controlled Trial Evaluating the Effectiveness of the Virtual Reality Distraction Compared to Current Practice, on Reducing Procedural Pain in Children and Adolescents Supported in Pediatric Onco-Hematology Unit	RCT, onco-hematological children or adolescent (8–17 years)	Device: Virtual Reality Headset. Procedures: various invasive procedures	Y	Y
NCT04092803	Recruiting	Estimated Primary Completion Date: June 2023 Estimated Study Completion Date: June 2024	Virtual Reality as a Distraction Technique for Performing Lumbar Punctures in Children and Young Adults With Leukemia: a Feasibility Study	Non-randomized Trial, leukemia patients, mixed population (10–25 years)	Device: Virtual Reality Headset. Procedures: lumbar puncture	Y	Y
NCT04934293	Recruiting	Estimated Primary Completion Date March 21, 2023	Virtual Reality for Children in Radiotherapy (REVER)	Cross-group cohort study, cancer patients (7–18 years)	Device: Virtual Reality Headset. Procedures: proton therapy	N	Y
NCT05042479	Not yet recruiting	Estimated Primary Completion Date 1 April 2022	Using of Virtual Reality to Relieve Procedural Pain in Pediatric Oncology (VIRTUOSO)	Clinical trial (within subject design), onco-hematological children or adolescent (7–18 years)	Device: Virtual Reality Headset. Procedures: various painful procedures	Y	Y
NCT05275881	Recruiting	Estimated Primary Completion Date: 4 April 2024	Impact of Virtual Reality in Pediatric Hematology and Oncology	RCT, onco-hematological children or adolescent (7–17 years)	Device: Virtual Reality Headset. Procedures: lumbar puncture and connection to an implantable chamber	Y	Y
NCT04931745	Recruiting	Estimated Primary Completion Date: 1 July 2023	Virtual Reality for Procedural Distress in Children Undergoing Port-a-Cath Access: A Randomized Controlled Trial	RCT, onco-hematological children or adolescent (5–17 years)	Device: Virtual Reality Headset. Procedures: PORT access	Y	N
NCT02995434	Unknown status	Estimated Primary Completion Date: 30 June 2021	Immersive Multimedia as an Adjunctive Measure for Pain Control in Cancer Patients	RCT, cancer patients, mixed population (>16 years)	Device: Virtual Reality Headset. Procedures: experience of chronic pain	Y	N
NCT04853303	Not yet recruiting	Estimated Primary Completion Date: 31 January 2025 Estimated Study Completion Date: 30 June 2025	The Use of a Virtual Reality Device (HypnoVR^®^) to Improve Chemotherapy-induced Nausea and Vomiting, Sleep Quality and Pain Among Children with Cancer in Hong Kong	RCT, cancer patients (9–18 years)	Device: Hypnosis VR. Procedures: experience of chemotherapy-induced nausea and vomiting, sleep quality, or pain.	Y	N
NCT04138095	Recruiting	Actual Primary Completion Date: 31 December 2020	Virtual Reality as an Adjunct to Management of Pain and Anxiety in Palliative Care	Clinical trial (within subject design), palliative patients (7–18 years)	Device: Oculus Quest Virtual Reality Headset. Procedures: Opioid and benzodiazepine use	Y	Y

N = no; Y = yes.

**Table 4 cancers-15-00985-t004:** Summary of the characteristics of included studies.

First Author, Year.	Country	Setting	Study Design	Population N. (Age), Gender	Cancer Diagnosis
Atzori et al., 2018 [50]	Italy.	A service of pediatric oncology and hematological diseases of a children’s hospital.	Within subjects crossover RCT.	n = 15 (7–17 years; M = 10.92, SD = 2.64), 66.7% males.	Unspecified diagnosis, oncology and hematology patients (73.3%) and other blood diseases.
Gerçeker et al., 2021 [47]	Turkey.	Pediatric hematology-oncology settings at two university hospitals.	RCT.	n = 42 (6–17 years; VR group: M = 11.2, SD = 3.1; control group: M = 11.7, SD = 3.7), 61.9% males.	Unspecified diagnosis, oncology (57.1%) and hematology (42.9%) patients.
Hundert et al., 2022 [48]	Canada.	A large metropolitan pediatric hematology/oncology outpatient clinic.	Pilot RCT.	n = 40 (8–18 years; VR group: M = 12.1, SD = 3.0; control group: M = 12.6, SD = 3.6), 63% males.	Leukemia (58%), brain tumor (25%), lymphoma (7%), other (10%).
Li et al., 2011. [52]	China.	A pediatric oncology unit at a largest acute-care hospital.	Quasi-experimental, non-equivalent control group, pretest-post-test, between-subject design (timing: at 7-day and 1-month pretest, post-test intervals).	n = 122 (8–16 years; VR group: M = 11.6, SD = 2.1; control group: M = 12.1, SD = 2.3). 53% males.	Leukemia (41%), lymphoma (25%), brain tumor (4%), germ-cell tumor (19%), osteosarcomas (11%).
Nilsson et al., 2009 [53]	Sweden.	A pediatric oncology unit at a children’s hospital.	Intervention-comparison group, parallel group design.	n = 42 (5–18 years; VR group: median = 11; control group: median = 11), 59.5% males.	Leukemia (23.8%), lymphoma (11.9%), CNS tumor (28.6), Other solid, tumor (21.4%), Hematological diseases (14.3%).
Russo et al., 2022 [54]	Italy.	An oncology-hematology department at a children’s hospital.	Experimental crossover design.	n = 22 (5–11 years; median = 8.4, range = 6.8–10.3), 72.7% males.	Unspecified diagnosis, cancer patients.
Sander Wint et al., 2002 [55]	USA.	A private, in-hospital treatment room, of a pediatric teaching hospital.	Experimental control group design.	n = 30 (10–19 years; VR group: median = 13.10, range = 9.90–18.70; control group: median = 14.30, range = 10.71–19.10), 53% males.	Acute lymphoblastic leukemia (67%), B-cell lymphoma (3%), lymphoma (3%), T-cell (7%), T-cell acute lymphoblastic leukemia (13%), T-cell lymphoma (7%).
Semerci et al., 2021 [51]	Turkey.	A pediatric oncology unit at a university hospital.	RCT, between-subject design.	n = 71 (7–18 years; VR group: M = 11.69, SD = 3.36; control group: M = 11.67, SD = 3.55), 51% males.	Unspecified diagnosis, cancer patients.
Sharifpour et al., 2020 [56]	Iran.	Three private chemotherapy clinics.	Quasi-experimental pre-test, post-test, between-subject design (timing: at 7-day and 1-month pretest, post-test intervals).	n = 30 (14–18 years; VR group: M = 14.8, SD = 2.4; control group: M = 15, SD = 1.85), gender percentage unspecified.	Osteosarcoma (43%), Ewing’s sarcoma (27%), brain tumor (7%), ovarian cancer (13%), skeletal muscle cancer (10%).
Schneider & Workman, 1999 [57]	USA.	An outpatient center of a comprehensive cancer center.	Interrupted time series with removed treatment.	n = 11 (10–17 years), 55% males.	Leukemia (64%), Hodgkin (36%).
Tennant et al., 2020 [45]	Australia.	The children’s cancer centre (ccc) at the royal children’s hospital (rch), melbourne,	Pilot RCT.	n = 90 (7–19 years; VR group: M = 11.59, SD = 3.61; control group: M = 11.6, SD = 2.77), 55.6% males.	Leukemia (44.4%), lymphoma (14.4%), brain tumor/CNS (4.4%), bone (17.8%), soft tissue (4.4%), melanoma (1%), germ cell (4.4%), other (8, 9%).
Wolitsky et al., 2005 [49]	USA.	A children’s hospital in a major metropolitan city.	RCT.	n = 20 (7–14 years; VR group: M = 11.20, SD = 2.25; control group: M = 9.80, SD = 2.30). 60% males.	Unspecified diagnosis, cancer patients.
Wong et al., 2021 [46]	China.	A children’s cancer centre of a regional public hospital.	RCT.	n = 108 (6–17 years; M = 10.4, SD = 3.6), 51.9% males.	Leukemia (75%), lymphoma (9.2%), bone tumor (10.2%), others (5.6%).

RCT = randomized controlled trial; M = mean; SD = standard deviation.

**Table 5 cancers-15-00985-t005:** Summary of the main results of included studies.

Ref	Procedure/ Application	Intervention/ VR Group (n.)	Control/ Comparison Group (n.)	Type of VR/ Equipment	Environment Software	Outcomes	Measures	Results
[50]	Venipuncture for chemotherapy, transfusions, magnetic resonance or blood analysis.	VR (when patients underwent the second venipuncture) (n = 15).	No distraction. Standard of care: when patients underwent the first venipuncture. Non-medical conversation by the nurse who performed the procedure.	**Immersive head-mounted display.** VR headset: helmet, earphones, and the personal 3D Viewer Sony (45 diagonal field of view, 1280 × 720 pixels per eye).	Snow World game: an icy canyon, where patients throw snowballs at penguins, snowmen, and other characters in VR, using a wireless mouse in hand not involved with the venipuncture.	**Pain: Y** Anxiety: N	Visual Analogue Scale (VAS) (scores range 0–10): (i) cognitive component—time spent thinking about pain); (ii) affective component—pain unpleasantness; (iii) sensory component—worst pain.	**Sig.: yes. Pain level:** “Time spent thinking about pain”: No-VR, M = 3.23 (SD = 2.98) vs. Yes-VR, M = 1.33 (SD = 1.05); *p* < 0.05. “Pain unpleasantness”: No-VR, M = 3.27 (SD = 3.43) vs. Yes-VR, M = 0.93 (SD = 1.16); *p* < 0.01. “Worst pain”: No-VR, M = 3.60 (SD = 3.00) vs. Yes-VR, M = 2.00 (SD = 1.20); *p* < 0.05.
[47]	Huber needle insertion into a subcutaneously implanted intravenous port for routine chemotherapy.	VR (when patients underwent the access to the port with a Huber needle). A single predetermined VR software (n = 21).	No distraction. Standard of care: staff delivered information to patients at least 1 h before the procedure using a standardized script (n = 21).	**Immersive head-mounted display.** VR glasses: Samsung Gear Oculus.	(I) “Ocean Rift”: swimming with marine animals underwater; (II) “Rilix VR”: riding a rollercoaster; (III) “In the eyes of animal”: exploring the forest through the eyes of woodland species.	**Pain: Y**	Wong–Baker FACES (WBF) Pain Rating Scale (scores range 0–10). The scale was used to assess: (I) patients self-report; (II) reports from the parents.	**Sig.: yes. Self-reported pain:** lower in the VR group than in the control group (M = 2.4, SD = 1.8; M = 5.3, SD = 1.8; *p* < 0.001, respectively). **Parent reported pain:** lower in the VR group than in the control group (M = 2.4, SD = 1.7; M = 5.1, SD = 2.0; *p* < 0.001, respectively).
						**Anxiety: Y**	The Children’s Anxiety Meter-State (CAM-S) (scores range 0–10).	**Sig.: yes. Self-reported anxiety:** lower in the VR group than in the control group (M = 2.9, SD = 2.0; M = 5.4, SD = 2.0; *p* < 0.001, respectively). **Parent reported anxiety:** lower in the VR group than in the control group (M = 2.9, SD = 2.0; M = 5.4, SD = 2.0; *p* < 0.001, respectively).
[48]	Huber needle insertion into a subcutaneously implanted intravenous port.	VR (auditory and visual stimuli for distraction), including topical anesthetic (adhesive anesthetic patches) (n = 20).	Non-immersive iPad distraction: patients watched a video while wearing the headphones. Standard of care: parental presence, unspecified topical anesthetics, child life specialist involvement (n = 20).	**Immersive head-mounted display.**	VR game/application: aiming rainbow balls at sea creatures as they explored an underwater environment in search of treasure.	**Pain: Y** Anxiety: N	11-point Numeric Rating Scale (NRS) (scores 0–10). The scale was used to assess: (I) patients self-report and reports from: (II) parents; (III) nurse; (IV) research staff.	**Sig.: N.A. Self-reported pain (pre)**: lower in iPad group (M = 0.3, SD = 0.7) then VR group (M = 0.9, SD = 1.6). **Self-reported pain (during):** lower in the VR group (M = 0.9, SD = 1.5) then in the iPad group (M = 1.3, SD = 2.3). **Parent reported pain (pre):** lower in the iPad group (M = 2.8, SD = 2.7) then VR group (M = 3.5, SD = 3.2). **Parent reported pain (during):** lower in the VR group (M = 1.6, SD = 2.4) then in the iPad group (M = 2.0, SD = 2.6). **Nurse reported pain (pre):** lower in the VR group (M = 1.7, SD = 1.6) then in the iPad group (M = 2.3, SD = 2.1). **Nurse reported pain (during):** lower in the VR group (M = 1.7, SD = 1.9) then in the iPad group (M = 2.9, SD = 2.7). **Research staff reported pain (pre):** lower in the VR group (M = 1.4, SD = 1.9) then in the iPad group (M = 1.6, SD = 2.4). **Research staff reported pain (during):** lower in the VR group (M = 1.3, SD = 1.6) then in the iPad group (M = 1.4, SD = 2.4).
[52]	Children with cancer admitted to the ward and undergoing active treatment.	VR: therapeutic play using virtual reality computer game. Post-test—phase 2 (one month of a washing period): all patients admitted to the unit in the phase 2 (n = 52).	No distraction. Standard of care: pre-test—phase 1: all patients admitted to the unit (n = 70).	**Non-immersive video projector.** PlayMotion system installed in a playroom. A device that transforms walls, floors, ceilings into wildly interactive playground (type of controller: children’s movements).	Projected play spaces: fly over a city, create trance-like waves, ripples, and colors to playing football, volleyball, or billiards.	**Anxiety: Y** Pain: N	Short form of State Anxiety Scale for Children (CSAS-C): 10 items rated on a 3-point Likert scale (total scores range 10–30; the higher the score, the higher the level of anxiety). The state anxiety levels were assessed on day 7 after admission for each group.	**Sig.: yes. Results of mixed between-within subjects ANOVA for** (i) changes in the scores of children in both groups across the two time periods: **main effect for time** (F = 33.03, *p* = 0.00, Eta squared = 0.21, Observed power = 0.96); (ii) changes in the scores of children over time depended on the types of **interventions: interaction effects** (F = 28.52, *p* = 0.00, Eta squared = 0.19, Observed power = 0.97). **Sig.: no. Results of between-subject effect:** difference in children’s scores between the two groups on day seven, **main effect for intervention** (F = 3.48, *p* = 0.07, Eta squared = 0.05, Observed power = 0.68). **No differences in children’s anxiety scores between the two groups on day 7.**
[53]	Venipunctures or Huber needle insertion into a subcutaneously implanted intravenous port.	VR + topical anesthetic (EMLA^®^ cream: n = 20; cold spray: n = 1) (n = 21).	No distraction. Standard of care: topical anesthetic (EMLA^®^ cream: n = 18; cold spray: n = 3) (n = 21).	**Non-immersive computer screen.** Virtual world games displayed on a standard personal computer with high-end consumer graphics card and a 3D display. GyroRemote remote control from Gyration.	‘‘The hunt of the diamonds’’ developed with Digital Content Creation software and Adobe Photoshop.	**Pain: Y** **Anxiety: Y**	**Self-reported pain:** Color Analogue Scale (CAS, range 0 -10; 0 = no pain, 10 = most pain) and Facial Affective Scale (FAS). **Observational pain:** The Face, Legs, Activity, Cry and Consolability Scale (FLACC, maximum total score = 10; higher score indicates more pain). **Self-reported anxiety/distress:** FAS.	**Sig.: no.** No statistically significant differences between the intervention group and the control group for CAS, and FLACC scales. The FLACC scores did not increase in the intervention group but increased significantly in the control group. After the procedure, the CAS (VR group 0.018–0.003 vs. control group. 0.001- 0.004); FAS (VR group 0.028- 0.008 vs. control group 0.028–0.001), and FLACC (VR group 0.163 -0.027 vs. control group 0.001–0.001) decreased significantly in both groups.
[54]	CVC dressing.	VR in 1 session.	No distraction: standard of care (during another session).	**Immersive head-mounted display.** VR headset: head-mounted-display device (Beha VR, Inc., Elizabethtown, KY, USA) and a tablet for clinicians’ control and supervision.	(I) “The MantaRay game”: peaceful underwater environment through navigation and control of manta-ray fish; (II) “The VitaminBee game”: throwing grains of pollen towards playful bees; (III) Diaphragmatic breathing exercise.	**Anxiety: Y** Pain: N	Revised Children’s Manifest Anxiety Scale (RCMAS-2) ( < 39 = no problematic level, 40–60 = normal level, 61–70 = moderately problematic level, >71 = extremely problematic level).	**Sig.: NA.** No variation in anxiety levels after VR intervention. **RCMAS-2 total score:** without VR = 36.5 (IQR = 32–48); with VR = 38.5 (IQR = 32–48).
[55]	Lumbar punctures (as a part of cancer therapy).	VR + topical anesthetic (EMLA^®^ cream, n = 14), conscious sedation (fentanyl and midazolam), parental presence, explanation of the procedure (n = 17).	No distraction. Standard of care: topical anesthetic (EMLA^®^ cream, n = 9), conscious sedation (fentanyl and midazolam), parental presence, explanation of the procedure (n = 13).	**Immersive head-mounted display.** VR glasses (i-O Display System LLC, Menlo Park, CA) with attached earphones, 3-D viewing of a 32 min long video.	“Escape” (Virtual i-O, Portland, OR): skiing down the Swiss Alps, a stroll down Paris sidewalks, visions of quiet mountain streams.	**Pain: Y** Anxiety: N	Visual Analogue Scale (VAS) (scores 0–100).	**Sig.: no.** Self-reported pain: lower in the VR group than in the control group (median = 7.0, range = 0.48; median = 9, range = 0–59; *p* = 0.77).
[51]	Port access procedure.	VR + standard of care (n = 35).	No distraction. Standard of care: parental presence (n = 36).	**Immersive head-mounted display.** VR headset: helmet, headphones, and the connected iPhone 6 mobile phone (The Piranha VR system).	“Rollercoaster video”: a roller coaster speeds up and slows down in a forest, accompanied by slow music.	**Pain: Y** Anxiety: N	Wong–Baker FACES (WBF) Pain Rating Scale (scores 0–10). The scale was used to assess: (I) patients self-report; (II) reports from the parents.	**Sig.: yes. Self-reported pain:** lower in the VR group than in the control group (M = 2.34, SD = 3.27; M = 5.02, SD = 3.35; *p* = 0.001, respectively). **Parent reported pain:** lower in the VR group than in the control group (M = 1.77, SD = 2.46; M = 4.67, SD = 2.56; *p* < 0.001, respectively).
[56]	Chemotherapy treatment.	VR intervention (n = 15).	No intervention (n = 15).	**Immersive head-mounted display.** VR headset (Samsung Gear VR, AAA VR Cinema v.1.6.1.—InstaVR) with Note 8 mobile device.	“Ocean journey”: a film of traveling along the beach and into the depths of the ocean.	**Pain: Y**	McGill Pain Questionnaire (MPQ) (a list of 78 words in 20 categories related to pain, scores 0–78)	**Sig.: yes. Results of between-subjects:** less perceived pain intensity in VR group compared to the control group (partial eta-squared = 0.90, *p* = 0.001). **Results of within-subjects:** significant time-by-group interaction effect (partial eta-squared = 0.37, *p* = 0.001). **Results of the estimated parameters of pain intensity in the stages of post-test, 7-day after and 1-month after:** significant differences between the VR group and the control group (*p* = 0.001).
						**Anxiety: Y**	Short version of the Pain Anxiety Symptom Scale (PASS-20) (scores 0–100).	**Sig.: yes. Results of between-subjects:** less symptoms in VR group compared to the control group (partial eta-squared = 0.95, *p* = 0.001). **Results of within-subjects:** significant time-by-group interaction effect (partial eta-squared = 0.59, *p* = 0.001). **Results of the estimated parameters of anxiety levels in the stages of post-test, 7-day after and 1-month after:** significant differences between the VR group and the control group (*p* = 0.001).
[57]	Chemotherapy treatment.	VR intervention during a single chemotherapy treatment (the second one) (n = 11).	No distraction. Standard of care (unspecified) during the first chemotherapy treatment.	**Immersive head-mounted display.** VR i-O headset connected with a personal computer.	“Magic Carpet”, “Sherlock Holmes Mystery”, “Seventh Guest”.	**Anxiety: Y** Pain: N	State-Trait Anxiety Inventory for Children (STAIC-1).	**STAIC-1—**Chemotherapy treatment (Chemo) 1, Chemo2, Chemo3: F = 2.47, *p* = 0.11. Results showed difference across the time period, but differences occurs at the first time measure and cannot be attributed to the VR.
[45]	Inpatient oncology admission: common cancer related emotional and physical distress symptoms.	VR (n = 61).	Non-immersive iPad distraction (Model A1475) and over ear headphones with content identical to the VR experience (n = 29).	**Immersive head-mounted display.** VR headset (Samsung Gear VR^®^ first-generation mobile HMD) and a smartphone (Galaxy S7^®^; Samsung); VRHMD and ear headphones, to ensure full immersion in a 3D, 270- degree of the virtual environment.	Three themes of VR experiences (10-min): simulated travel to Australian national parks, Australian zoos, and global city tourist spots (i.e., ‘Travel’ experience).	**Pain: Y**	Visual Analogue Scale (VAS) (scores range 0–10).	**Sig.: no. Pain level:** changes across VAS measures post-treatment when compared to iPad control; VR group (M = 10.97, SD = 11.23), iPad group (M = 12.82, SD = 11.34) (mean difference = 1.85, *p* = 0.475). **Moderation analysis:** lower pain scores in the VR group (m = 8.19, se = 2.36) compared to the iPad group (m = 15.52, se = 2.92, *p* = 0.056) for females.
						**Anxiety: Y**	Visual Analogue Scale (VAS) (scores 0–10).	**Sig.: no. Anxiety:** changes across VAS measures post-treatment when compared to iPad control; VR group (M = 10.20, SD = 15.26), iPad group (M = 11.60, SD = 15.44) (mean difference = 1.40, *p* = 0.692). **Moderation analysis:** lower state anxiety in the VR group (m = 5.51, se = 3.02) compared to the iPad group (m = 13.99, se = 3.75, *p* = 0.083) for females.
[49]	Port access procedure.	VR (n = 10).	No distraction (n = 10).	**Immersive head-mounted display.** VR head-mounted display, noise-cancelling headphones (for sound and a connected joystick controller (for interaction).	“The gorilla habitat at Zoo Atlanta”.	**Pain: Y**	**Self-reported pain:** Visual Analogue Scale (VAS) (scores range 0–100). **Observational pain (observed from a researcher):** The Children’s Hospital of Eastern Ontario (CHEOPS) (scores: six behavioral categories scored from 0 = no pain to 3 = severe pain).	**Sig.: no. Composite score of self-reported measures** (mean of VAS anxiety and pain scores) during the procedure, retrospective recorded after the procedure: **VR intervention** M = 12.00, SD = 16.36; **Non-VR intervention** M = 34.45, SD = 41.80; t (18) =3.03, *p* = 0.10. **Sig.: yes**. **CHEOPS** (during the procedure): **VR intervention** M = 4.90, SD = 0.99; **Non-VR intervention** M = 8.30, SD = 2.41; t (18) =4.13, *p* < 0.01.
						**Anxiety: Y**	**Anxiety:** Visual Analogue Scale (VAS) (scores 0–100). **How-I-Feel questionnaire**, before the procedure (20 questions on a 3-point Likert scale).	
[46]	Peripheral intravenous cannulation (PIC).	VR in patients underwent the first attempt for a PIC (no additional attempt of PIC was considered) (n = 54).	No distraction. Standard of care: non-medical conversation by phlebotomists who performed the procedure (explanation and verbally comfort) (n = 54).	**Immersive head-mounted display.** VR headset: Google cardboard goggles.	2 animated videos from “Minions” (visual and auditory stimuli requiring minimal movement of the head).	**Pain: Y**	Visual Analogue Scale (VAS) (scores range 0–10).	**Sig.: yes.** (Generalized estimating equation (GEE) model to assess difference across time between the two groups). **Pain level—**VR group: a reduction in pain after PIC than the control group (estimated mean difference = −1.69, *p* = 0.007). **Sensitivity analysis** (VR effect for reduction in pain stratified by age groups): 12–17 years, estimated mean difference = −2.20 (*p* = 0.034); 6–11 years, estimated mean difference = −1.38 (*p* = 0.077).
						**Anxiety: Y**	Short form of the State Anxiety Scale for Children (CSAS-C): 10 items rated on a 3-point Likert scale (total scores range 10–30; the higher the score, the higher the level of anxiety).	**Sig.: yes. Anxiety level:** the VR group showed a reduction in anxiety level after PIC than the control group (estimated mean difference = −3.50, *p* < 0.001). **Sensitivity analysis** (VR effect for reduction in anxiety stratified by age groups): 12–17 years, estimated mean difference = −2.90 (*p* = 0.008); 6–11 years, estimated mean difference = −3.85 (*p* < 0.001).

RCT = randomized controlled trial; M = mean; SD = standard deviation; N = no; Y = yes; N.A. = not applicable; Sig. = significance; VR = virtual reality.

**Table 6 cancers-15-00985-t006:** Other outcomes measured across studies.

Reference	Other Outcomes Measured	Results/Conclusion
**Studies measured as primary outcome: PAIN**		
[50]	**(i) Quality of the VR experience (presence and realism)** (2 ad hoc questions). **(ii) Fun** (1 ad hoc question). **(iii) Length of the procedure** (minutes).	**Quality of the VR experience:** mean presence score = 7.39 (strong sense of going inside the virtual world), mean realism of VR objectives = 6.80 (moderately real). **Fun:** No-VR, M= 2.93 (SD = 3.58) vs. Yes-VR, M = 8.80, SD = 1.42, *p* < 0.0001. **Total time of the procedure:** Yes-VR, M = 3.09 vs. NO-VR, M = 4.45 (*p* > 0.05).
[48]	**(i) Fear** (The Child Fear Scale (CFS)). **(ii) Distress** (11-point NRS) (scores 0–10). **(iii) Immersiveness** (12 items on a 3-point scale) (aggregated scores (0–24). **(iv) Pain catastrophizing** (6-item state version of the Pain Catastrophizing Scale for Children (PCS-C) and for parents (PCS-P)) (scores 0–60). **(v) Patients, nurses, and parents’ satisfaction and acceptability** (questions on a 4-point scale from “not at all” to “very much”).	**Fear:** similar scores in both groups, 63% (n = 12) of VR participants and 67% (n = 12) of iPad participants reported no fear. **Distress:** 80% of VR participants (n = 16) reported no distress, compared to 56% of iPad participants (n = 10). **Immersiveness:** in the VR group was significantly higher (M = 16.4, SD = 5.4) than in the iPad group (M = 14.5, SD = 5.4) (*p* = 0.0318). **Pain catastrophizing:** lower in the VR group (M = 9.0, SD =11.5) than in the iPad group (M = 13.8, SD =14.9). **Participant-reported, parent-reported, and nurse-reported acceptability** was high in both study groups, as well as **satisfaction**: high satisfaction with the VR intervention (children, parent, and nurses). Nurses found use of the VR headset integrated well into their workflow.
[53]	**(i) Pulse rate** (pulse-oxygen monitor). **(ii) Examination of patients’ response to the use of VR equipment** during the procedure (semi-structured qualitative interviews).	**Pulse rate:** 5 min before and during the procedure; no significant difference in pulse rate. **Patients’ response to the use of VR equipment:** - 13/21 expressed satisfaction with the game and equipment. - Interviews—qualitative analyses: (1) The VR game should correspond to the child and the medical procedure; (2) non-immersive VR is a positive experience for children undergoing a minor procedure such as venous puncture or a subcutaneous venous port access: although the children sometimes saw a 3D effect, they did not think it was necessary for the distraction; (3) children enjoyed the VR game and found that it did distract them during the procedure.
[55]	**(i) Subjects’ evaluation of the VR experience and the effectiveness of VR glasses as a distraction technique** (semi-structured interview, 10 item with a combination of open-ended questions and response set questions).	- 77% of participants in the VR group said the VR glasses helped to distract them from the procedure. - 94% of participants said they wanted to use the VR glasses again during their next lumbar puncture.
[51]	None reported.	-
**Studies measured as primary outcome: ANXIETY**		
[52]	**(i) Children’s depressive symptoms** [The Center for Epidemiologic Studies Depression Scale for Children (CES-DC)].	Children in the VR group reported statistically significant fewer depressive symptoms than children in the control group on day 7.
[54]	**(i) Distress level** (The Distress Thermometer). **(ii) Children’s and parent’s satisfaction** (an ad-hoc developed questionnaire). (**iii) Perceptions of health care workers (HCW) about the applicability of VR** (interviews).	**Distress:** the comparison of distress levels after medication by VR yielded a strong decrease in median scores (without VR = 4 (IQR 3–5); with VR = 2 (IQR 0–2). **Satisfaction:** high rates of overall satisfaction. 5% of families and children reported some discomfort in the use of the device. **HCWs’ Opinion:** 7/8 found VR useful in clinical practice; 2/8 questioned the usability of the VR device.
[57]	**(i) Distress** (The Symptom Distress Scale) (13 items on a Likert Scale, scores 13–65).	VR could mitigate chemotherapy-related symptom distress (*p* < 0.10). The lowest mean SDS score occurred immediately following the second chemotherapy treatment (Chemo 2), after the use of VR. SDS at Chemo1, Chemo2, Chemo3: F = 3.30, *p* = 0.06. SDS values were high prior to chemotherapy administration, dropped immediately following chemotherapy treatment and rebounded at the 48 h post-chemotherapy measure.
**Studies measured as primary outcomes: PAIN and ANXIETY**		
[47]	**(i) Fear (patients’ self-reported and parent reported)** (The Child Fear Scale (CFS)) (scores 0–4).	**Self-reported fear:** lower in the VR group than in the control group (M = 0.8, SD = 0.9; M = 2.0, SD = 1.0; *p* < 0.001, respectively). **Parent reported fear:** lower in the VR group than in the control group (M = 0.9, SD = 0.9; M = 2.0, SD = 1.0; *p* < 0.001, respectively).
[56]	**(i) Patients’ confidence with chronic pain** (Pain self-efficacy questionnaire (PSEQ)) (10 items on a 7-point Likert scale, scores 0–60). **(ii) Pain catastrophizing** (The Pain Catastrophizing Scale (PCS)) (13 items on a 5-point Likert scale, scores 0–52).	VR intervention explained the 70.1% of the variation of PSEQ and the 82.4% of the PCS levels (*p* = 0.001). **Results of the estimated parameters of patients’ confidence in the stages of post-test, 7-day after and 1-month after:** significant differences between the VR group and the control group (*p* = 0.001).
[45]	**Primary outcomes:** (i) Child state, (ii) Positive mood, (iii) Anger, (iv) Nausea (child report—Visual Analogue Scale (VAS), 100 cm horizontal lines). (v) Pulse rate (pulse-oxygen monitor). **Secondary outcomes:** (i) Trait Anxiety (child report—Spence Children’s Anxiety Scale (SCAS) short form). (ii) Child illness status (parent proxy report, Pediatric Quality of Life Inventory™ Cancer Module (PedsQL Cancer Module)). (iii) Immersion (child report Adapted version of the Total Immersion subscale of the Augmented Reality Immersion (ARI) questionnaire). (iv) Enjoyment (child report VAS Enjoyment thermometer). (v) Simulator sickness (child report, Child Simulation Sickness Questionnaire (CSSQ)).	**Primary outcomes:** increasing in overall child emotional well-being for both VR and iPad intervention, with no statistically significant differences (*p* > 0.05) between the two groups (improvements in scores across all subjective wellbeing Measures, i.e., enhanced positive mood state; reductions in anxiety and anger; and lowered nausea and pain perception). **Immersion and enjoyment with intervention:** participants in the VR condition reported slightly greater immersion and enjoyment compared to iPad, but these differences were not statistically significant.
[49]	**(i) Pulse rate** (pulse-oxygen monitor). **(ii) Examination of patients’ experience, thoughts, and feelings** (semi-structured interviews).	**Pulse rate:** during the procedure, there were significant differences between groups: VR group M = 95.80, SD = 19.3; control group M = 117.60, SD = 25.70; *p*< 0.05. **Patients’ experience:** children in the VR group recalled significantly more actions in their narratives, elaborated more, and tended to mention more thoughts and emotions, than children in the control group.
[46]	**(i) Pulse rate** (pulse-oxygen monitor). **(ii) Length of procedure** (minutes).	**Patients’ pulse rate** (T1 = during the procedure, T2 = immediately after the procedure): T1 estimated mean difference = 1.69 (*p* = 0.34); T2 estimated mean difference = 1.19 (*p* = 0.51). **Length of PIC procedure:** mean duration significantly shorter among the VR group, M = 2.70, SD = 0.74 than the control group, M = 3.41, SD = 2.13; estimated mean difference = −0.75 (*p* = 0.017).

## Data Availability

The study did not report any data.

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
