# Peer review of "Immersive and Non-Immersive Virtual Reality for Pain and Anxiety Management in Pediatric Patients with Hematological or Solid Cancer: A Systematic Review"

_cancers, 2023, doi:10.3390/cancers15030985_

Round 1
Reviewer 1 Report
this is a comprehensive systematic review on the use of virtual reality for pain and anxiety management in pediatric patients.
. Introduction :
-can further improve with studies on pediatric patients.
- the difference between immersive and non-immersive, is that so much difference, if yes, need more details in the background
-
Methods:
-comprehensive using PRISMA
-the difference between immersive and non-immersive, is that so much difference, if yes, need more details in the methods
results & discussion:
-can be better to report immersive and non-immersive to pain and anxiety
and then, draw the conclusion
thanks
Author Response
This is a comprehensive systematic review on the use of virtual reality for pain and anxiety management in pediatric patients.
We thank the reviewer for his/her appreciation. We answered step by step all the reviewer’s suggestions, hoping to have well interpreted his/her requests. All changes are reported in red in the text.
. Introduction :
-can further improve with studies on pediatric patients.
We revised and added recent literature on pediatrics, as suggested by the reviewer.
- the difference between immersive and non-immersive, is that so much difference, if yes, need more details in the background
We are glad for this helpful comment, and we added more details as requested. We would like to underline that as a systematic review without meta-analysis, we tried to synthesize literature of both types of interventions (immersive and non-immersive) in order to have a comprehensive view of the topic.
Methods:
-comprehensive using PRISMA
We appreciate this comment.
-the difference between immersive and non-immersive, is that so much difference, if yes, need more details in the methods
As requested, we added a more detailed explanation related to our choice to include both interventions and in the description of the inclusion criteria (Intervention’s included).
results & discussion:
-can be better to report immersive and non-immersive to pain and anxiety and then, draw the conclusion
As suggested, we revised and added details in the Result section: summary of results, by considering the difference between non immersive VR and immersive VR for both pain and anxiety.
Reviewer 2 Report
This paper is a systematic review for immersive and non-immersive virtual reality for pain and anxiety management in pediatric patients with hematological or solid cancer. It is interesting and well written. It is difficult to make comparisons among studies because the content of each intervention is so different. And painful interventions also differed between studies (cerebrospinal fluid puncture and rooting seem to differ slightly.)In addition, as written in the limitation, the immersive virtual reality seems to be quite different in the content and construction among studies, it is not clear which element of the content was effective. (Is it the scenario of contents , the reality of the images, or the cuteness of the characters, etc.?) However, I think that the manuscript is well organized,Author Response
This paper is a systematic review for immersive and non-immersive virtual reality for pain and anxiety management in pediatric patients with hematological or solid cancer. It is interesting and well written. It is difficult to make comparisons among studies because the content of each intervention is so different. And painful interventions also differed between studies (cerebrospinal fluid puncture and rooting seem to differ slightly.)In addition, as written in the limitation, the immersive virtual reality seems to be quite different in the content and construction among studies, it is not clear which element of the content was effective. (Is it the scenario of contents , the reality of the images, or the cuteness of the characters, etc.?) However, I think that the manuscript is well organized,
We are glad that the Reviewer appreciated our review, we are conscious that some limits are present, and we take the opportunity to add some of the suggested limits in the revised Ms. We thank the Reviewer for this precious insight.